# In Vitro Antiplatelet Activity of Mulberroside C through the Up-Regulation of Cyclic Nucleotide Signaling Pathways and Down-Regulation of Phosphoproteins

**DOI:** 10.3390/genes12071024

**Published:** 2021-06-30

**Authors:** Hyuk-Woo Kwon, Dong-Ha Lee, Man Hee Rhee, Jung-Hae Shin

**Affiliations:** 1Department of Biomedical Laboratory Science, Far East University, Eumseong 27601, Korea; kwonhw@kdu.ac.kr; 2Department of Biomedical Laboratory Science, Namseoul University, Cheonan 31020, Korea; dhlee@nsu.ac.kr; 3Molecular Diagnostics Research Institute, Namseoul University, Cheonan 31020, Korea; 4Laboratory of Physiology and Cell Signaling, College of Veterinary Medicine, Kyungpook National University, Daegu 41566, Korea; rheemh@knu.ac.kr; 5Department of Biomedical Laboratory Science, Catholic Kwandong University, Gangneung 25601, Korea

**Keywords:** mulberroside C, intracellular calcium, glycoprotein IIb/IIIa, granule secretion, clot retraction

## Abstract

Physiological agonists trigger signaling cascades, called “inside-out signaling”, and activated platelets facilitate adhesion, shape change, granule release, and structural change of glycoprotein IIb/IIIa (αIIb/β3). Activated αIIb/β3 interacts with fibrinogen and begins second signaling cascades called “outside-in signaling”. These two signaling pathways can lead to hemostasis or thrombosis. Thrombosis can occur in arterial and venous blood vessels and is a major medical problem. Platelet-mediated thrombosis is a major cause of cardiovascular disease (CVD). Therefore, controlling platelet activity is important for platelet-mediated thrombosis and cardiovascular diseases. In this study, focus on *Morus Alba* Linn, a popular medicinal plant, to inhibit the function of platelets and found the containing component mulberroside C. We examine the effect of mulberroside C on the regulation of phosphoproteins, platelet-activating factors, and binding molecules. Agonist-induced human platelet aggregation is dose-dependently inhibited by mulberroside C without cytotoxicity, and it decreased Ca^2+^ mobilization and p-selectin expression through the upregulation of inositol 1, 4, 5-triphosphate receptor I (Ser^1756^), and downregulation of extracellular signal-regulated kinase (ERK). In addition, mulberroside C inhibited thromboxane A_2_ production, fibrinogen binding, and clot retraction. Our results show antiplatelet effects and antithrombus formation of mulberroside C in human platelets. Thus, we confirm that mulberroside C could be a potential phytochemical for the prevention of thrombosis-mediated CVDs.

## 1. Introduction

Many studies have been conducted on health and longevity, and new agents that are beneficial for human health can be challenging to identify. Among various candidates, medicinal plants are major constituents of alternative medicines that lead to new drugs [1]. Cardiovascular diseases (CVDs), in particular, affect human health. Various factors have been identified as causative of cardiovascular disease; however, the role of platelets is putatively very important. Platelets play a fundamental role in thrombosis, and the available antiplatelet drugs effectively reduce thrombosis in patients with CVDs. However, some side effects of these drugs have been reported [2]. Therefore, to develop new antiplatelet drugs, we focused on the antiplatelet activity of various medicinal plants.

*M. Alba* Linn, a popular medicinal plant belonging to the family Moraceae, has long been used in traditional medicine in many Asian countries [3]. Regarding the effects of *M. Alba* extracts for reducing thrombosis, it has been reported that the *M. Alba* leaves extract inhibits rat platelets and showed synergistic effects with *Schisandra chinensis* [4,5]. In addition, morusinol isolated from *M. Alba* inhibits agonist-induced rabbit platelets [6]. Therefore, we examined the antiplatelet effects of mulberroside C isolated from *M. Alba* [7].

When endothelial walls are injured, collagen can bind to integrin α2β1 and glycoprotein VI of platelets. These interactions lead to the inside-out signaling pathways. During platelet activation, phospholipase Cγ_2_ breaks down phosphatidylinositol 4,5-bisphosphate into inositol diacylglycerol and 1,4,5-trisphosphate (IP_3_), and released IP_3_ binds to the IP_3_ receptor on the endoplasmic reticulum [8]. These signaling pathways facilitate Ca^2+^ mobilization, α-and δ-granule secretion, and platelet aggregation. The p-selectin is located inside the α-granule, and during inside-out signaling, p-selectin is re-expressed on the platelet surface through α-granule release [9]. Another mechanism that regulates calcium is an influx from extracellular spaces. Depletion of Ca^2+^ in the endoplasmic reticulum facilitates the influx of extracellular Ca^2+^, which is controlled by extracellular signal-regulated kinases [10]. 

Another platelet activator is thromboxane A_2_ (TXA_2_). The mitogen-activated protein kinase p38 (p38^MAPK^) is an essential mediator for the activation of cytosolic phospholipase A_2_ (cPLA_2_) through phosphorylation at Ser^505^ [11]. Activated cPLA_2_ hydrolyzes the plasma membrane and releases arachidonic acid, which is synthesized as TXA_2_ by cyclooxygenase and TXA_2_ synthase [12]. TXA_2_ acts as an autacoid that activates other platelets and facilitates thrombosis [13].

The most well-known substances for the inhibitory action of platelets are cyclic nucleotides, cyclic adenosine monophosphate (cAMP), and cyclic guanosine monophosphate (cGMP) [14]. In normal blood circulation, endothelial cells release nitric oxide and prostaglandin I_2_, which maintain the inactive platelet condition. Two platelet-inactivating substances act on platelets, increasing cAMP and cGMP levels. Vasodilator-stimulated phosphoprotein (VASP) and inositol 1,4,5-triphosphate receptor type I (IP_3_RI) are major targets of cAMP and cGMP. These signaling molecules can affect αIIb/β_3_ activity and Ca^2+^ mobilization in platelets [15]. Therefore, medical plants that regulate cyclic nucleotides can be used for platelet-mediated thrombosis. 

In this study, we examined whether mulberroside C inhibits platelet aggregation and thrombus formation through the concentration of cyclic nucleotides and associated signaling molecules.

## 2. Materials and Methods

### 2.1. Materials

Mulberroside C (Figure 1) was provided by ChemFaces (Wuhan, China). Mouse monoclonal to CD62P (P-selectin) antibody was purchased from Biolegend (San Diego, CA, USA). Fura 2-AM (2-acetoxymethyl) and Alexa Fluor 488-conjugated fibrinogen were obtained from Invitrogen (Eugene, OR, USA). The BCA protein assay kit was obtained from Pierce Biotechnology (Rockford, IL, USA). The serotonin ELISA kit was provided by Labor Diagnostika Nord GmbH and Co. (Nordhorn, Germany). Collagen, U46619, and thrombin were purchased from Chrono-Log Co. (Havertown, PA, USA). The thromboxane B_2_ assay kit, cAMP, cGMP enzyme immunoassay kit, and ATP assay kit were obtained from Cayman Chemical (Ann Arbor, MI, USA). Cell signaling (Beverly, MA, USA) supplied antiphospho-p38^MAPK^, antiphospho-ERK (1/2), antiphospho-VASP (Ser^157^), antiphospho-VASP (Ser^239^), antiphospho-cPLA_2_ (Ser^505^), antiphospho-PI3K (Tyr^458^), antiphospho-Akt (Ser^473^), antiphospho-inositol-3-phosphate receptor type I (Ser^1756^), antiphospho-PLC_γ__2_ (Tyr^759^), anti-β-actin, and antirabbit secondary antibodies.

### 2.2. Preparation of Suspension of Human Platelets

Human platelet-rich plasma (PRP) was obtained from the Korean Red Cross Blood Center (Suwon, Korea). Experiments are conducted on human PRP isolated from 4 independent PRP probes. The platelets were then washed twice with washing buffer (138 mM NaCl, 2.7 mM KCl, 12 mM NaHCO_3_, 0.36 mM NaH_2_PO_4_, 5.5 mM glucose, and 1 mM Na_2_EDTA), and resuspended in suspension buffer (138 mM NaCl, 2.7 mM KCl, 12 mM NaHCO_3_, 0.36 mM NaH_2_PO_4_, 0.49 mM MgCl_2_, 5.5 mM glucose, 0.25% gelatin). All experiments were approved by the Catholic Kwandong University Institutional Review Board (CKU-20-01-0109). The platelet suspension was adjusted to a concentration of 5 × 10^8^/mL cells/mL according to a previous study [16].

### 2.3. In Vitro Platelet Aggregation Assay

Platelet suspensions (10^8^/mL) were preincubated for 2 min with mulberroside C (50–150 μM), and then agonists (collagen, thrombin, and U46619) are added for aggregation. Dimethyl sulfoxide (DMSO) is used to dissolve mulberroside C, and the final concentration of DMSO is 0.1% (not interfere with the platelet aggregation). Platelet aggregation was measured for five minutes at 37 °C under stirring conditions. An aggregometer (Chrono-Log, Havertown, PA, USA) is a device for measuring light transmission. The platelet suspension is turbid, but becomes transparent when platelets are aggregated. Therefore, we analyze the change in light transmission, and the change in the light transmission was reported as the aggregation rate. 

### 2.4. Cytotoxicity Analysis

A lactate dehydrogenase leakage assay was conducted for cytotoxicity. Human platelets (10^8^/mL) were incubated with mulberroside C (50–150 μM) for 20 min and centrifuged at 12,000× *g*. Lactate dehydrogenase was detected in the supernatant using an ELISA reader (TECAN, Salzburg, Austria).

### 2.5. Ca^2+^ Mobilization Analysis

The Fura 2-AM (5 μM) and PRP mixture were preincubated at 37 °C for 60 min, and then the platelets (10^8^/mL) were washed with washing buffer as previously described. After the washing step, platelets were suspended in suspension buffer and preincubated with or without G-derrone for 3 min at 37 °C. The platelets were stimulated with collagen (2.5 μg/mL) in the presence of 2 mM CaCl_2_. A spectrofluorometer (Hitachi F-2700, Tokyo, Japan) was used to measure Fura 2-AM fluorescence according to the Grynkiewicz method [17] to calculate the Ca^2+^ values.

### 2.6. Serotonin and ATP Analysis

Platelet aggregation was conducted with mulberroside C (50–150 μM) for 6 min at 37 °C. The reactant is transferred to a new centrifuge tube and briefly centrifuged at 500× *g*, and the supernatant was used for serotonin and ATP detection. Serotonin was detected using an ELISA reader (TECAN, Salzburg, Austria), and ATP release was measured with a Synergy HT Microplate Reader (BioTek Instruments, Winoosku, VT, USA).

### 2.7. Flow Cytometric Determination of P-Selectin Release

Human platelet suspension (10^8^/mL) was preincubated with mulberroside C (50–150 μM) and then stimulated with collagen (2.5 μg/mL). After platelet aggregation, the reaction tube was incubated with Alexa Fluor 488 antihuman CD62P under dark conditions. Next, the platelets were washed twice with ice-cold PBS and fixed with 0.5% paraformaldehyde. The p-selectin expression was measured using flow cytometry and CellQuest software (BD Biosciences, San Diego, CA, USA).

### 2.8. Thromboxane B_2_ Analysis

Because thromboxane A_2_ (TXA_2_) is quickly converted to thromboxane B_2_ (TXB_2_), TXA_2_ generation was measured by TXB_2_. After platelet aggregation with mulberroside C (50–150 μM), the reaction tube was terminated by adding indomethacin (0.2 mM) and centrifugation of the supernatant. TXB_2_ was detected using an ELISA reader (Tecan, Salzburg, Austria).

### 2.9. Immunoblotting Assay

Platelet aggregation was lysed by adding lysis buffer. The platelet lysates were measured using a BCA protein assay kit (Pierce Biotechnology, IL, USA). After electrophoresis (8% SDS-PAGE), proteins (12 μg) were transferred onto membranes and treated with primary (1:1000) and secondary antibodies (1:10,000). Western blotting analysis was conducted using the Quantity One program (Bio-Rad, Hercules, CA, USA).

### 2.10. Flow Cytometric Determination of αIIb/β3 Activation

For the experiment of fibrinogen binding to αIIb/β3, platelet aggregation was conducted with mulberroside C (50–150 μM) for 5 min. After aggregation, the reaction tube was incubated with fibrinogen (Alexa Fluor 488-conjugated) for 15 min and fixed with 0.5% paraformaldehyde. Since αIIb/β3 of activated platelets binds very strongly to fibrinogen, fluorescent fibrinogen was used as an indicator for the fibrinogen binding experiment. Flow cytometry was used to measure binding (BD Biosciences, San Jose, CA, USA).

### 2.11. Measurement of cAMP and cGMP

Washed human platelets (10^8^/mL) were preincubated with mulberroside C (50–150 μM) for 2 min in the presence of 2 mM CaCl_2_, then stimulated with collagen (2.5 μg/mL) for 5 min at 37 °C for platelet aggregation. The aggregation was terminated by the addition of 80% ice-cold ethanol. The reactant is transferred to a new centrifuge tube and briefly centrifuged at 500× *g*, and the supernatant was used for cAMP and cGMP detection. cAMP and cGMP were measured using EIA kit with ELISA reader (TECAN, Salzburg, Austria).

### 2.12. Fibrin Clot Retraction

For the fibrin clot retraction experiment, platelets were reacted with thrombin (0.05 U/mL). Human platelet-rich plasma (300 μL) was incubated with mulberroside C (50–150 μM) for 30 min at 37 °C, and the clot reaction was triggered by thrombin. After reacting for 15 min, pictures of fibrin clots were taken using a digital camera. Image J (v1.46) was used to convert the clot area (National Institutes of Health, Bethesda, MD, USA).

### 2.13. Data Analyses

All data are presented as the mean ± standard deviation with various numbers of observations. To determine major differences among groups, analysis of variance was performed, followed by the Tukey-Kramer method. SPSS 21.0.0.0 software (SPSS, Chicago, IL, USA) was used for statistical analysis, and *p* < 0.05 was considered statistically significant.

## 3. Results

### 3.1. Mulberroside C Blocks Platelet Aggregaion and Cytotoxicity

Platelet suspensions (10^8^/mL) were preincubated for 2 min with mulberroside C (50–150 μM, Figure 1). Three agonists (collagen, thrombin, U46619) are used for platelet aggregation assay. Mulberroside C inhibited all agonist-induced platelet aggregation (Figure 2A–C) without cytotoxicity (Figure 2D); among them, collagen-induced platelets treated with mulberroside C (50, 75, 100, and 150 μM) were the most strongly inhibited (17.9%, 43.9%, 81.8% and 96.2%). Half-maximal inhibitory concentration (IC_50_) is the most widely used measure of a drug’s efficacy. The IC_50_ of mulberroside C was 77.3 in collagen-induced human platelet (Figure 2E). 

### 3.2. Mulberroside C Blocks Ca^2+^ Levels and IP_3_RI-, ERK-Phosphorylation

Next, we investigated the phosphorylation of calcium mobilization. As shown in Figure 3A, calcium level ([Ca^2+^]_i_) in cytosol was elevated to 98.2 ± 0.5 nM by collagen (2.5 μg/mL). However, mulberroside C (50–150 μM) reduced the collagen-induced increase in [Ca^2+^]_i_ levels (Figure 3A). It has been reported that if cAMP/cGMP-dependent kinases phosphorylate inositol 1, 4, 5-triphosphate receptor type I (IP_3_RI), [Ca^2+^]_i_ mobilization is inhibited [14]. Another pathway is Ca^2+^ influx from extracellular spaces, which increases the [Ca^2+^]_i_ level. Depletion of the intracellular Ca^2+^ level in the endoplasmic reticulum by collagen is known to trigger Ca^2+^ influx, which is facilitated by ERK [10]. Therefore, we investigated whether these two signaling molecules and mulberroside C (50 to 150 μM) increased IP_3_RI phosphorylation and decreased ERK phosphorylation (Figure 3B,C).

### 3.3. Mulberroside C Blocks P-Selectin Expression and Serotonin Release

Next, we investigated whether mulberroside C is involved in granule release in human platelets. Increased calcium level in cytosol activates myosin light chain and pleckstrin phosphorylation to trigger granule release (δ-granule and α-granule). Because mulberroside C decreased [Ca^2+^]_i_ levels through IP_3_RI phosphorylation and ERK dephosphorylation, we investigated granule release. As shown in Figure 4A, collagen elevated the expression of p-selectin (Figure 4A,B); however, mulberroside C (50–150 μM) inhibited collagen-induced p-selectin expression in a dose-dependent manner (Figure 4A,B). Next, we also determined whether δ-granule release was regulated by mulberroside C. As shown in Figure 4C,D, collagen increased serotonin release, and ATP release was suppressed by mulberroside C (50–150 μM).

### 3.4. Mulberroside C Blocks Thromboxane B_2_ Production, and Dephosphorylation of cPLA_2_, p38^MAPK^

TXA_2_ stimulates platelet activation increased TXA_2_ may play a role in the pathogenesis of myocardial infarction, stroke, and atherosclerosis. As shown in Figure 5A, collagen (2.5 μg/mL) stimulation elevated the TXA_2_ concentration to 66.2 ± 1.8 ng/10^8^ platelets. However, mulberroside C (50–150 μM) inhibited TXA_2_ production dose-dependently (Figure 5A). To identify the inhibitory action of mulberroside C on TXA_2_ production, we investigated cPLA_2_ and mitogen-activated protein kinase p38 (p38^MAPK^). cPLA_2_ hydrolyzes arachidonic acid from the membrane, and the released arachidonic acid transforms to TXA_2_. During the transformation, the p38^MAPK^ elevates the enzyme activity of cPLA_2_. As shown in Figure 5B,C, mulberroside C suppressed cPLA_2_ and p38^MAPK^ phosphorylation in a dose-dependent manner. 

### 3.5. Mulberroside C Blocks Platelet Binding to Fibrinogen by Limiting αIIb/β3 Affinity and VASP, PI3K/Akt Dephosphoryation

αIIb/β3 is essential for platelet adhesion, binding, and spreading. Collagen-induced fibrinogen binding to αIIb/β3, and its binding rate was 93.7 ± 3.9% (Figure 6A). However, mulberroside C significantly decreased conformational change of αIIb/β3 BA (Figure 5B and Figure 6A(c–f)). Next, we investigated αIIb/β3 associated signaling molecules, vasodilator-stimulated phosphoprotein (VASP), and PI3K/Akt. VASP regulates actin, but its phosphorylation by kinases suppressed the αIIb/β3 affinity [18,19]. Furthermore, phosphoinositide 3-kinase (PI3K) and Akt are the most important mediators in human platelets, leading to adhesive function, platelet spreading, and αIIb/β_3_ activation [20,21]. Our data showed that mulberroside C significantly upregulated VASP phosphorylation (Ser^157^, Ser^239^) (Figure 6C,D) and downregulated PI3K/Akt phosphorylation (Figure 6E,F).

### 3.6. Mulberroside C Elevated Cyclic Nucleotides and Limits Thrombin-Induced Clot Retraction and Phospholipase C_γ2_ Phosphorylation

Next, we investigated cyclic adenosine monophosphate (cAMP) and cyclic guanosine monophosphate (cGMP) concentration in human platelets. As shown in Figure 7A,B, mulberroside C significantly increased cAMP and cGMP levels and more strongly increased cGMP levels than cAMP levels. Taken together, we investigated whether mulberroside C can affect fibrin clot retraction. As shown in Figure 7C, thrombin-stimulated PRP was contracted, and the clot is formed. However, mulberroside C (50–150 μM) effectively delayed clot formation (Figure 7D). Activated αIIbβ3 triggers tyrosine phosphorylation of β3 integrin tail and activates phospholipase C_γ2_ (PLC_γ2_). PLC_γ2_ has been reported to be essential for platelet spreading and clot retraction [22]. As shown in Figure 7C, collagen elevated PLC_γ2_ phosphorylation was suppressed by mulberroside C (Figure 7E).

## 4. Discussion

*M. Alba* Linn (*M. Alba*), known as white mulberry, belongs to the family Moraceae and is commonly regarded as a food source for silkworms. *M. Alba* is widespread throughout East Asia and is used in traditional remedies as a diuretic, antiheadache, expectorant, and antidiabetic agent, and almost all parts of the plant are used in traditional medicine. Several compounds have been isolated from various parts of *M. Alba*, including bark, fruits, and leaves [3]. With regards to cardiovascular diseases, administration of *M. Alba* (500 mg/kg body weight) has shown cardioprotective effects in myocardial infarcted rats [23] and vascular protection effects [24]. In prior research investigating the effects of *M. Alba* effects on platelets, *M. Alba* extract inhibited in vitro rat platelet aggregation [4,5], and morusinol isolated from *M. Alba* inhibited in vitro rabbit platelet aggregation, thromboxane B_2_ formation, and in vivo ferric chloride-induced thrombosis [6]. Because the *M. Alba* extract showed antiplatelet effects, we searched for a new candidate and confirmed that mulberroside C has antiplatelet effects. 

Mulberroside C suppressed collagen-, thrombin-, and U46619 (TXA_2_ analog) induced human platelet aggregation in a dose-dependent manner (Figure 2A–C). Next, we confirmed that mulberroside C suppressed [Ca^2+^]_i_ levels. [Ca^2+^]_i_ levels are regulated by Ca^2+^ mobilization from the endoplasmic reticulum and Ca^2+^ influx from the extracellular space. It is well known that Ca^2+^ mobilization is activated by collagen and is inhibited by IP_3_RI phosphorylation. In addition, ERK phosphorylation has been reported to affect calcium influx. Therefore, we investigated whether mulberroside C inhibited [Ca^2+^]_i_ levels and confirmed that mulberroside C can suppress [Ca^2+^]_i_ levels through the phosphorylation of IP_3_RI and dephosphorylation of ERK (Figure 3B,C) and inhibited p-selectin expression, serotonin release, and ATP release (Figure 4A,C,D).

Next, we determined that mulberroside C suppressed TXA_2_ release (Figure 4A). TXA_2_ acts as a strong agonist, leading to hemostasis and thrombosis. We focused on the regulatory signaling molecules of TXA_2_, such as cPLA_2_ and p38^MAPK^. Mulberroside C inhibited p38^MAPK^ and cPLA_2_ phosphorylation (Figure 5B,C).

On the platelet surface, αIIb/β_3_ is the most important receptor for binding with fibrinogen and fibrin for hemostasis and thrombosis [25]. We investigated whether mulberroside C suppresses conformational change of αIIb/β3 by regulation of the VASP (Figure 6A,B). Mulberroside C suppressed αIIb/β3 affinity and increased VASP phosphorylation (Ser^157^ and Ser^239^). Intracellular cAMP and cGMP are strong negative molecules that are regulated by enzymes, such as cyclic adenylate/guanylate cyclase and phosphodiesterases. These cyclic nucleotides inhibit αIIb/β3 affinity and Ca^2+^ mobilization. In our study, mulberroside C increased cAMP and cGMP levels (Figure 4D,E), and these cyclic nucleotides can elevate the phosphorylation of VASP (Ser^157^ and Ser^239^) and IP_3_RI (Ser^1756^). PI3K and Akt are the most important mediators in human platelets, leading to adhesive function, filopodia formation, platelet spreading, and αIIb/β_3_ activation [20,21]. Mulberroside C inhibited PI3K p85 (Tyr^458^) and Akt (Ser^473^) phosphorylation (Figure 6C,D) and increased cAMP and cGMP levels (Figure 7A,B). αIIb/β3 with fibrin is involved in fibrin clot formation, which plays an important role in thrombosis and hemostasis. The clot forms a clot plaque and gradually contracts formation. The contraction process is caused by the interaction between fibrin and αIIb/β3, and the internal signaling molecules of platelets. Therefore, we investigated whether mulberroside C affects thrombin-induced fibrin clot retraction. As shown in Figure 7C, mulberroside C strongly suppressed the retraction. Our study had some limitations, in that it was conducted in vitro and did not consider other factors in vivo. Moreover, since our study is not an in vivo study thus, it is difficult to prove its effectiveness on the human body. To resolve these questions, in vitro whole blood aggregation, animal tests (in vivo, ex vivo), and clinical trials in humans should be accompanied. However, since the present study did not deal with these experiments, it must be conducted in a future study to prove the true antiplatelet effect of mulberroside C. However, based on the in vitro study, we suggest that mulberroside C has potential as a drug.

This study found that mulberroside C decreases calcium mobilization, fibrinogen binding to αIIb/β_3_, fibronectin adhesion, and thrombin-facilitated clot retraction by regulating the associated signaling molecules, such as IP_3_RI, ERK, cPLA_2_, p38, VASP, PI3K/Akt, and PLC_γ2_. Therefore, we suggest that mulberroside C from *M. Alba* would be a useful compound for preventing thrombosis.

## Figures and Tables

**Figure 1 genes-12-01024-f001:**
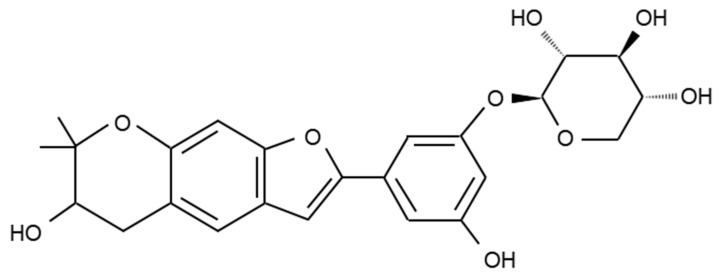
Chemical structure of mulberroside C (MW. 458.46) isolated from M. Alba.

**Figure 2 genes-12-01024-f002:**
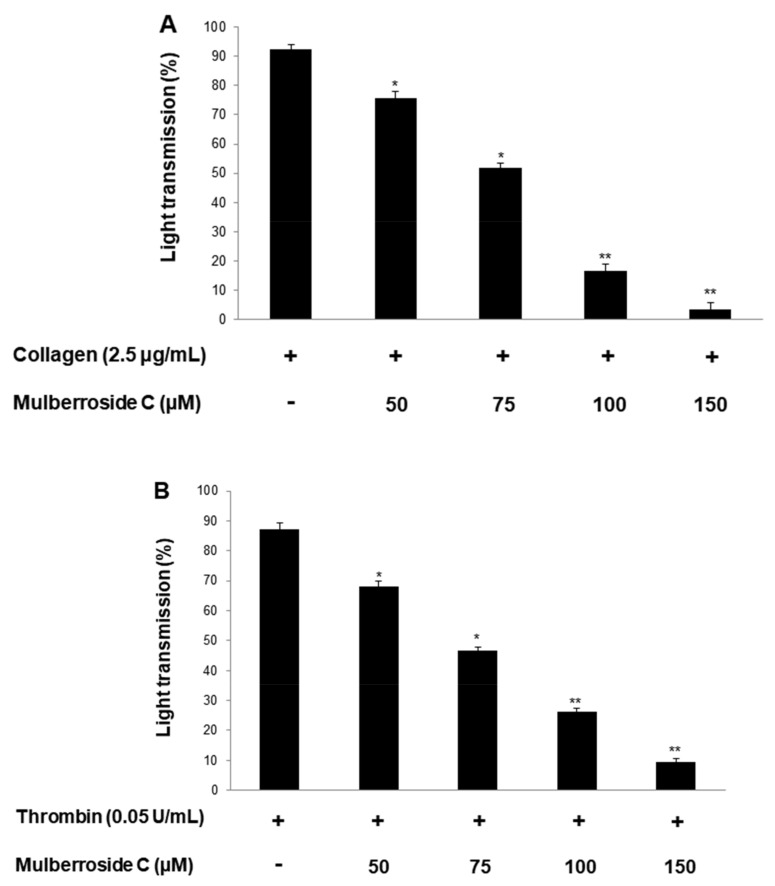
Effects of mulberroside C on platelet aggregation, half-maximal inhibitory concentration, and cytotoxicity (**A**) Effect of mulberroside C on collagen-induced human platelet aggregation; (**B**) Effect of mulberroside C on thrombin-induced human platelet aggregation; (**C**) Effect of mulberroside C on U46619-induced human platelet aggregation; (**D**) Effect of mulberroside C on cytotoxicity; (**E**) Half-maximal inhibitory concentration (IC_50_) value of mulberroside C in collagen-induced human platelet aggregation. Platelet aggregation and cytotoxicity were carried out as described in Section 2. The data are expressed as the mean ± standard deviation (*n* = 4). * *p* < 0.05, ** *p* < 0.01 versus each agonist-stimulated human platelets. NS, not significant.

**Figure 3 genes-12-01024-f003:**
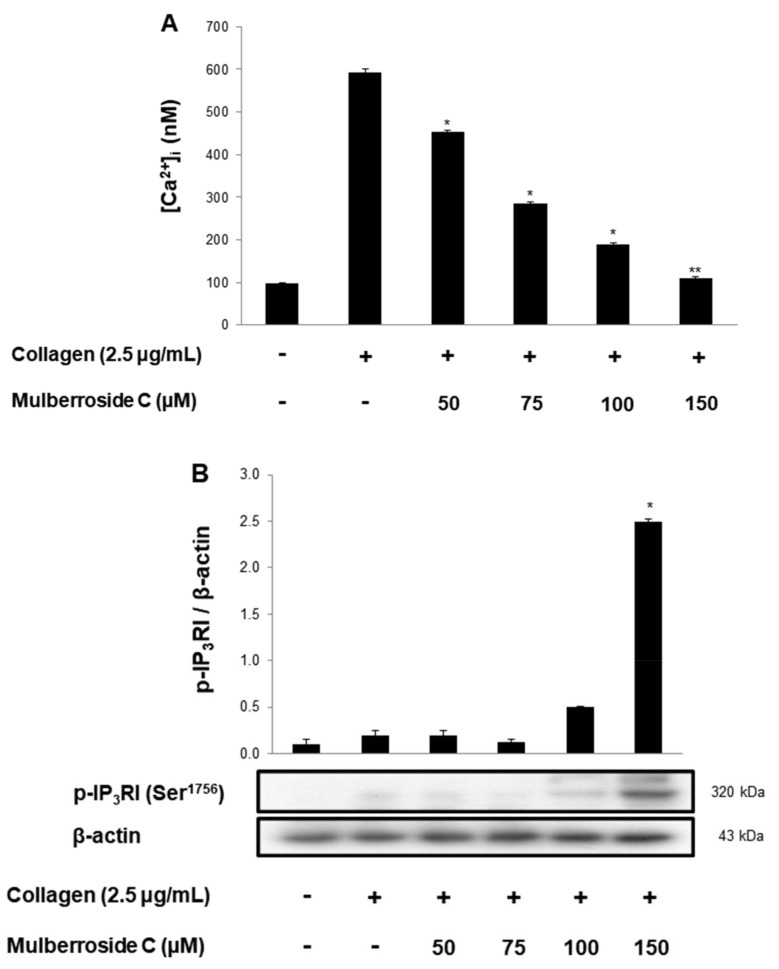
Effects of mulberroside C on Ca^2+^ mobilization, IP_3_RI and ERK phosphorylation (**A**) Effect of mulberroside C on collagen-induced Ca^2+^ mobilization; (**B**) Effect of mulberroside C on collagen-induced IP_3_RI phosphorylation; (**C**) Effect of mulberroside C on collagen-induced ERK phosphorylation. Measurement of Ca^2+^ mobilization, Western blot was performed as described in Section 2. The data are expressed as the mean ± standard deviation (*n* = 4). * *p* < 0.05, ** *p* < 0.01 versus the collagen-stimulated human platelets.

**Figure 4 genes-12-01024-f004:**
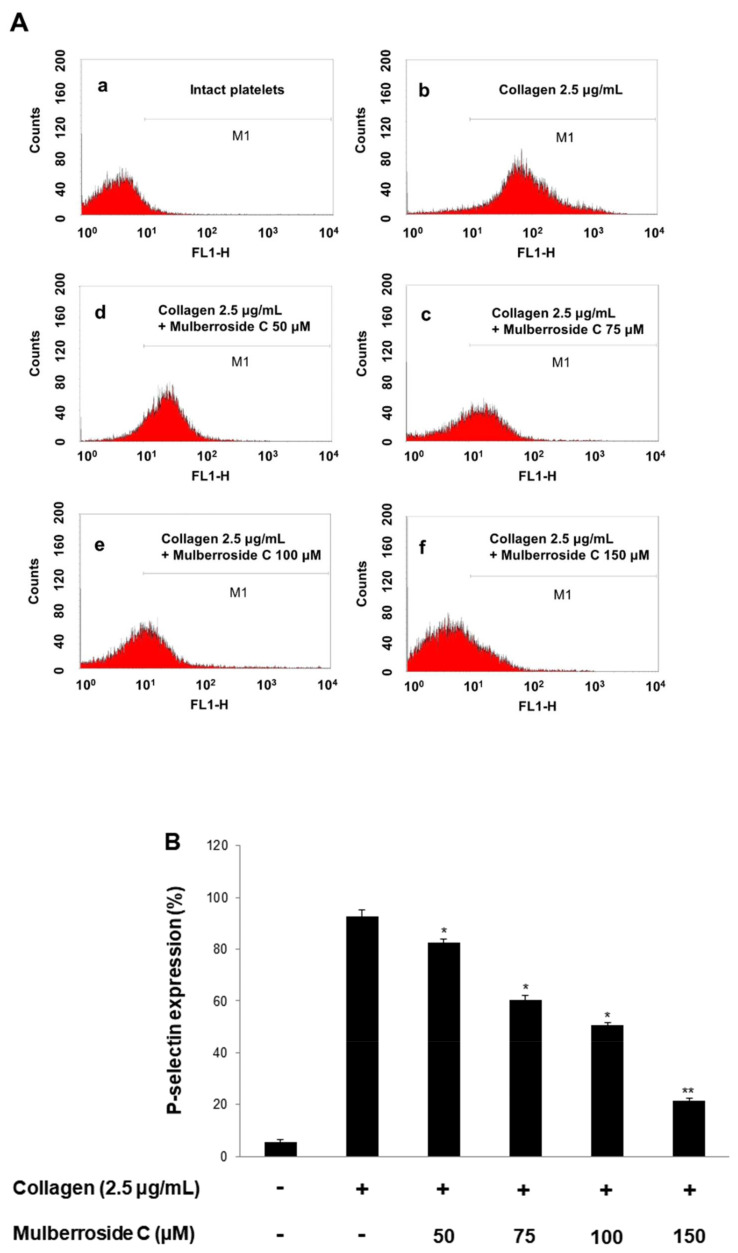
Effects of mulberroside C on p-selectin expression, serotonin, ATP release. (**A**) The flow cytometry histograms on p-selectin expression; (**B**) Effects of mulberroside C on collagen-induced p-selectin expression (%); (**C**) Effects of mulberroside C on collagen-induced serotonin release; (**D**) Effects of mulberroside C on collagen-induced ATP release. Determination of p-selectin expression, serotonin, ATP was carried out as described in Section 2. The data are expressed as the mean ± standard deviation (*n* = 4). * *p* < 0.05, ** *p* < 0.01 versus the collagen-stimulated human platelets.

**Figure 5 genes-12-01024-f005:**
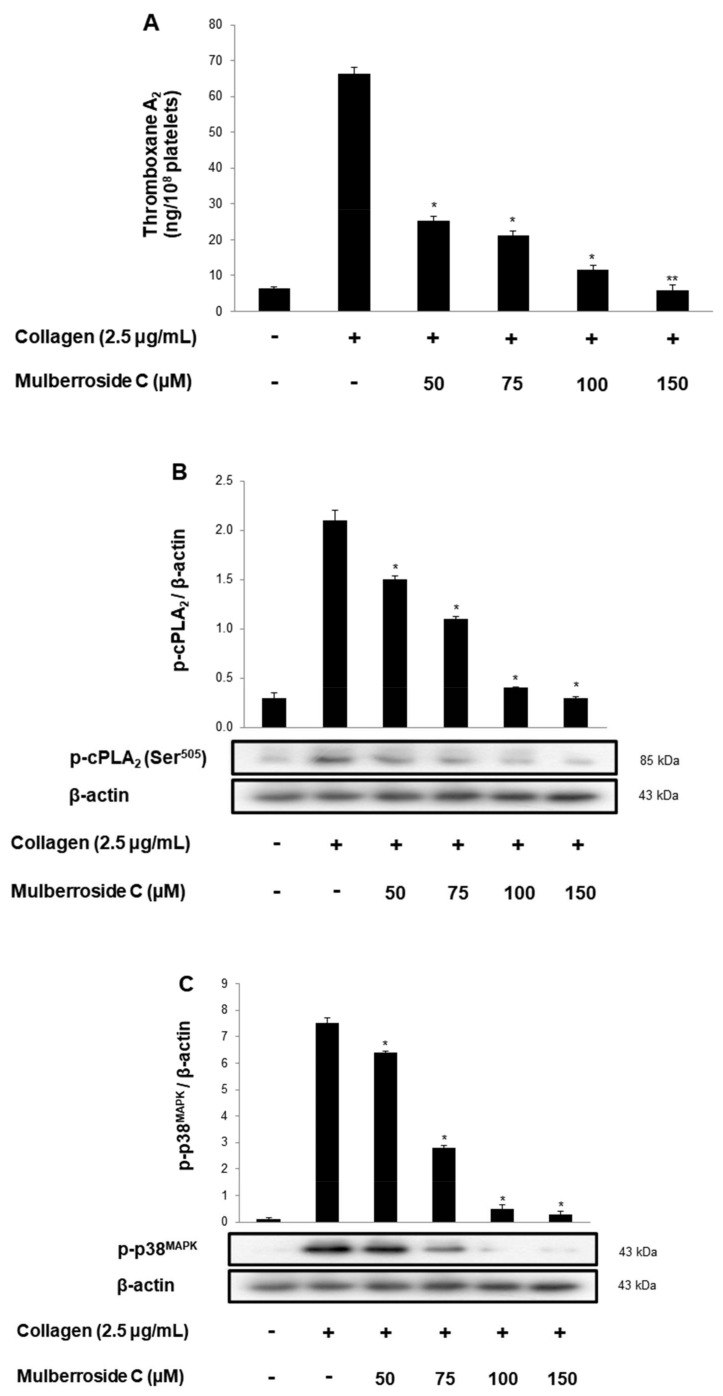
Effects of mulberroside C on TXA_2_ production, cPLA_2_, p38 phosphorylation (**A**) Effects of mulberroside C on collagen-induced TXA_2_ generation; (**B**) Effect of mulberroside C on collagen-induced cPLA_2_ phosphorylation; (**C**) Effect of mulberroside C on collagen-induced p38 phosphorylation. Measurement of TXA_2_ generation and Western blot was performed as described in Section 2. The data are expressed as the mean ± standard deviation (*n* = 4). * *p* < 0.05, ** *p* < 0.01 versus the collagen-stimulated human platelets.

**Figure 6 genes-12-01024-f006:**
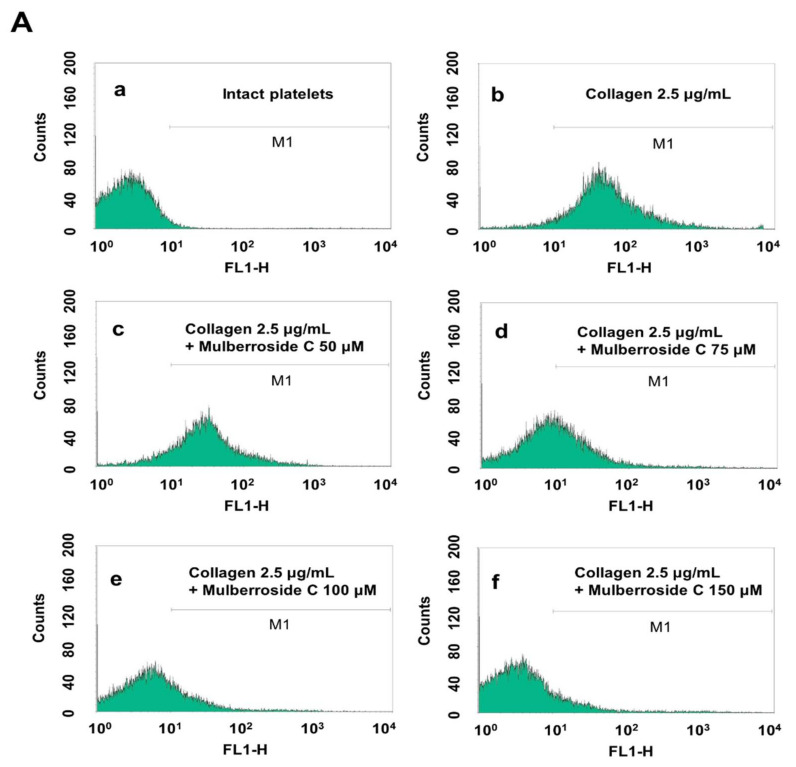
Effects of mulberroside C on fibrinogen binding to αIIb/β3, VASP, and PI3K/Akt phosphorylation (**A**) The flow cytometry histograms on fibrinogen binding; (**B**) Effects of mulberroside C on collagen-induced fibrinogen binding (%); (**C**) Effect of mulberroside C on collagen-induced VASP (Ser^157^) phosphorylation; (**D**) Effect of mulberroside C on collagen-induced VASP (Ser^239^) phosphorylation; (**E**) Effect of mulberroside C on collagen-induced PI3K (Tyr^458^) phosphorylation; (**F**) Effect of mulberroside C on collagen-induced Akt (Ser^473^) phosphorylation. Measurement of fibrinogen binding, fibronectin adhesion, and Western blot was carried out as described in Section 2. The data are expressed as the mean ± standard deviation (*n* = 4). * *p* < 0.05, ** *p*< 0.01 versus the collagen-stimulated human platelets.

**Figure 7 genes-12-01024-f007:**
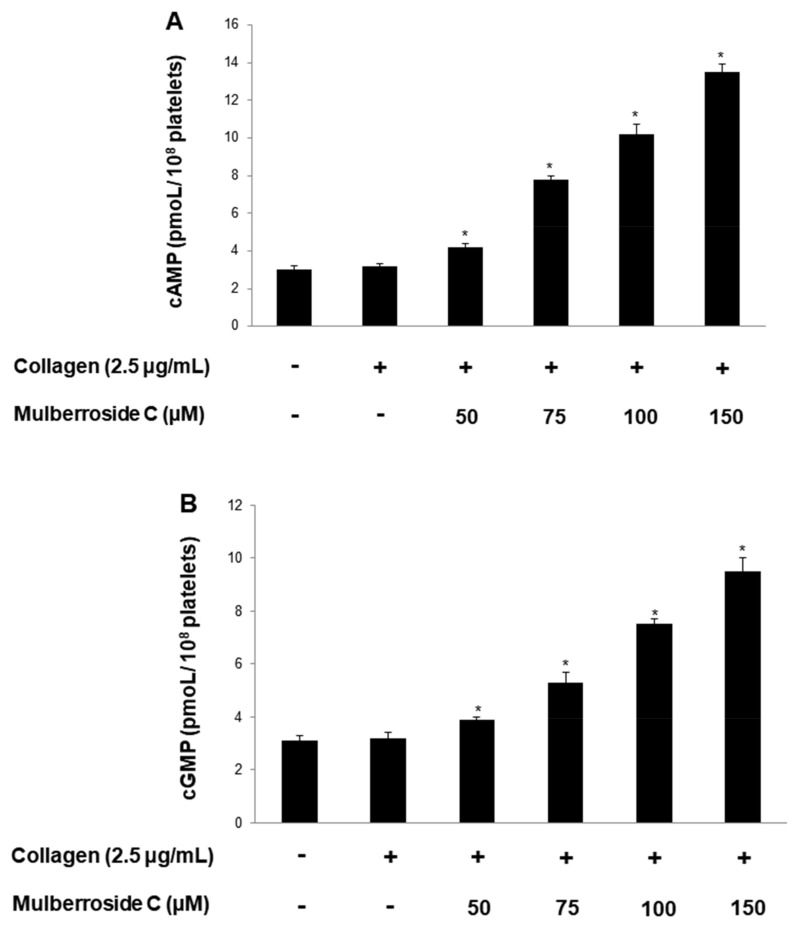
Effects of mulberroside C on cAMP and cGMP concentration, fibrin clot retraction, and PLCγ2 phosphorylation; (**A**) Effect of mulberroside C on collagen-induced cAMP production; (**B**) Effect of mulberroside C on collagen-induced cGMP production; (**C**) Photographs of fibrin clot; (**D**) Effects of mulberroside C on thrombin-retracted fibrin clot (%); (**E**) Effect of mulberroside C on collagen-induced PLCγ2 (Tyr759) phosphorylation. Quantification of fibrin clot retraction and Western blot, cAMP and cGMP was performed as describe in Section 2. The data are expressed as the mean ± standard deviation (*n* = 4). * *p* < 0.05 versus the unstimulated human PRP, # *p* < 0.05 versus the thrombin-stimulated human PRP. † *p* < 0.05 versus the colla-gen-stimulated human platelets.

## Data Availability

The data presented in this study are available on request from the corresponding author. The data are not publicly available due to privacy.

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
