# Peer review of "In Vitro Antiplatelet Activity of Mulberroside C through the Up-Regulation of Cyclic Nucleotide Signaling Pathways and Down-Regulation of Phosphoproteins"

_genes, 2021, doi:10.3390/genes12071024_

Round 1

Reviewer 1 Report

The manuscript describes antiplatelet activity of mulberroside  C, and potential molecular mechanisms of inhibition of platelet function by mulberroside  C . This is a model in vitro study using human platelets. The novelty of the study is that Authors reported,  first time, inhibitory effect of mulberroside  C isolated from Morus alba on agonist-induced platelet reactivity from human. This report is a good complement to knowledge about the antiplatelet properties of Morus alba  extracts and its compounds.

The aim and the results of the study are interesting, however the paper in the present form, in my opinion, cannot be recommended for publication – major revision is required.

In my opinion, the model study reported in this manuscript can be named “preliminary study”. Although, antiplatelet activity of mulberroside  C was shown for a few markers of platelet function (platelet aggregation, P selectin expression, calcium mobilization, thromboxane A2 production, fibrinogen binding), the inhibitory effects were observed only for isolated platelets from human PRP from blood center. Additionally, the measurements were repeated only 4 times (n=4), and it is not clear that platelets were isolated from 4 independent PRP probes. The experiments in whole human blood should be done (platelet aggregation with agonists like collagen; P selectin expression or fibrinogen binding to platelets) to confirm platelet inhibition in the presence mulberroside  C. The measurements in whole blood are closer to physiological conditions and the Authors should explain why the experiments were done only for isolated platelets.

Minor comments:

What was the control sample? The isolated platelet suspension with DMSO?

The data should be analyzed with using paired tests, and also non-parametric (because n=4).

There is no Figure 1 in the manuscript.

The all results from this study should be concluded in the end of Discussion section.

Reviewer 2 Report

 Hyuk-Woo Kwon et al. report new results concerning the actions of mulberroside c, a bioactive substance extracted from Morus alba. They described some significant effects of mulberroside c: inhibition of collagen-, thrombin-, and U46619-induced platelet aggregation, suppression of intracellular calcium release via the phosphorylation of IP3RI and dephosphorylation of ERK; suppression of p-selectin expression, serotonin release, ATP release, and further, TXA2 secretion, along with p38MAPK and cPLA2 phosphorylation. The authors obtained suggestive results for the reduction of αIIb/β3 affinity for fibrinogen, mulberroside c, thus exerting a complex anti-aggregant effect. This is well-designed research, with results yet not reported.

I have only three remarks for the improvement of the manuscript:

  1. The expression of GP IIb/IIIa on the platelet surface is known to be constitutive. However, mulberroside c is a novel anti-aggregant, and I think that checking GP IIb/IIIa expression by direct fluorescent antibody labeling would have been informative. The authors should shortly explain their design.
  2. The described effects intracellular and inside-out signaling, and thus, a synthetic explanatory figure of these pathways would help to understand for the readers.
  3. A final language editing is needed, as some slight misspellings occur, like in subtitles 3.1. “Mulberroside C blocks platelet aggregaion and cytotoxicity” and 3.5., “Mulberroside C blocks platelet binding to fibrinogen by limiting αIIb/β3 affinity and VASP, 252 PI3K/Akt dephosphoryation”.
